# Mind the Gender Gap in Marine Recreational Fisheries

Pablo Pita [1,*], Gillian Barbara Ainsworth [1], Bernardino Alba [2], Josep Alós [3], José Beiro [4], Pablo Martín-Sosa [5], Llibori Martínez [6], Begoña Marugán-Pintos [7], Beatriz Morales-Nin [3], Estanis Mugerza [8], Beatriz Nieto [9], Javier Seijo [10], Marta Pujol [11], Ana Tubío [1], Leonardo A. Venerus [12] and Sebastian Villasante [1]

1   CRETUS, Faculty of Business Administration and Management, University of Santiago de Compostela, 15705 Santiago de Compostela, Spain; gill.ainsworth@usc.es (G.B.A.); ana.tubio.gomez@usc.es (A.T.); sebastian.villasante@usc.es (S.V.)
2   Alianza de Pesca Española Recreativa Sostenible, 07300 Palma, Spain; bernadi.alba@gmail.com
3   Instituto Mediterráneo de Estudios Avanzados, IMEDEA (CSIC-UIB), 07190 Esporles, Spain; alos@imedea.uib-csic.es (J.A.); beatriz@imedea.uib-csic.es (B.M.-N.)
4   Federación Gallega de Pesca Marítima Responsable y Náutica de Recreo, 36310 Vigo, Spain; beirodiz@icloud.com
5   Centro Oceanográfico de Canarias, CSIC-IEO, 38180 Santa Cruz de Tenerife, Spain; pablo.martin-sosa@ieo.csic.es
6   International Forum for Sustainable Underwater Activities, 08028 Barcelona, Spain; llibori66@gmail.com
7   Facultad de CC Jurídicas y Sociales, Universidad Carlos III, 28911 Madrid, Spain; bmarugan@polsoc.uc3m.es
8   AZTI, Marine Research, Basque Research and Technology Alliance (BRTA), 48395 Sukarrieta, Spain; emugerza@azti.es
9   WWF España, 28005 Madrid, Spain; bnieto@wwf.es
10  Department of Political Science and Sociology, University of Santiago de Compostela, 15705 Santiago de Compostela, Spain; javierseijo.villamizar@usc.es
11  Catalan Institute for Ocean Governance Research (ICATMAR–Direcció General de Política Marítima i Pesca Sostenible), 08003 Barcelona, Spain; martap@gencat.cat
12  Centro para el Estudio de Sistemas Marinos, Consejo Nacional de Investigaciones Científicas y Técnicas (CESIMAR–CONICET), Puerto Madryn 9120, Argentina; leo@cenpat-conicet.gob.ar
*   Correspondence: pablo.pita@usc.es

**Abstract:** One of the most relevant information gaps in worldwide fisheries is related to the origin and consequences of the gender gap. Recreational fisheries show a remarkable gender gap, which has been especially poorly addressed in the scientific literature. In 2021, the Spanish Working Group on Marine Recreational Fishing (MRF) developed a broad diagnosis on the participation of women in MRF and agreed on a roadmap to address negative impacts derived from the gender gap. The network experts concluded that there is an urgent need to include the gender gap in the agendas of scientists, fishery managers, policy-makers, stakeholder organizations, and civil society. There is a need to better understand the gender-related socio-ecological impacts of MRF to improve fisheries governance and to develop policies and initiatives that facilitate the full access of women to the benefits derived from the practice of MRF. Establishing economic incentives, increasing the visibility of female success references, developing fishing-related programs specifically designed for girls, and promoting the perception of MRF as a family leisure activity in contact with nature will increase women's engagement.

**Keywords:** fisherwomen; recreational fishing; gender gap; social inequalities; outdoor activities; well-being

## 1. Introduction

Recreational fishing is one of the most practiced leisure activities in the global seas, with millions of practitioners who contribute through their expenses and investments to the blue economy [1]. There is also a flourishing tourism industry based on fishing guide services and charter boat rentals in coastal countries in the Atlantic [2,3], Indian [4,5],

and Pacific oceans [6,7]. The contribution of marine recreational fishing (MRF) to social welfare also includes non-market benefits generated by recreational fishing experiences [8]. In addition, benefits for the health and well-being of recreational fishers have been described [9], mainly derived from the reduction in stress and from seafood-rich diets, with direct implications for public health services [10]. On the other hand, negative impacts include the additional pressure exerted on fish stocks, already affected by commercial fleets in many cases [11,12], and alterations derived from fishers' disturbance to ecosystems [13].

Although most recreational fishers are men, the socio-ecological repercussions of the remarkable gender gap in MRF have been little addressed in the scientific literature [14,15]. The existence of a gender gap is not exclusive to MRF but affects different leisure activities carried out outdoors [16]. Consequently, gender inequalities in the distribution of benefits from outdoor recreation have already been described [17]. Some negative consequences of the gender gap in commercial fisheries have also been described [18], including not only explicit violence and the social exclusion of fisherwomen [19] but also inadequate fisheries management procedures [20,21] that result in lower incomes for fisherwomen [22].

During 2021, the Spanish Working Group on MRF (GT PMR—its acronym in Spanish), a network created to reinforce the socio-ecological sustainability of MRF, designed different activities to identify and address urgent challenges to research and the governance of MRF. Among these challenges, it included an analysis of the participation of women in MRF due to the lack of specific information on its potential consequences, as already described for commercial fisheries [18]. The aim of the GT PMR is to foster collaboration and knowledge transfer between fisher organizations (11 MRF associations operating at national and regional levels), research centers (18 units from 12 scientific institutions), companies, NGOs (2 organizations operating at both national and international levels), and the central (from 2 different ministries) and regional public administrations (from all 9 coastal autonomous regions). In October 2021, the GT PMR organized a virtual workshop that included participatory sessions in which the network developed a broad diagnosis on the participation of women in MRF and agreed on a roadmap to address the potential negative effects derived from the gender gap, as an opportunity to overcome the hegemonic masculinity in outdoor sports culture [16]. The results of the workshop have been summarized in this work to promote scientific debate on the topic and to guide institutional activities and public policies in the future.

## 2. Methods

The dynamics of the GT PMR activities developed in 2021 to analyze the participation of women in MRF was agreed upon by a core group of six network members (four females and two males), including four academics (two females) and one female representative, each from a commercial fisherwomen's association and an environmental NGO (ENGO). In October 2021, the core group, helped by professional facilitators with previous experience in running participatory sessions in the fisheries sector, organized a two-hour workshop for GT PMR members to develop dialogue on the participation of women in MRF. During the workshop, available information on the extent of the gender gap in MRF (including scientific papers), challenges faced by fisherwomen's organizations, and the potential consequences for society of the existence of gender gaps in leisure activities was presented by the women of the network with previous knowledge and experience in these fields, including published scientific papers. Subsequently, the attendees participated in a dynamic session designed to foster participation separately in three sectoral groups (scientists, fishers, and civil society), where proposals were made to reduce the gender gap in MRF, which were then agreed upon in a plenary debate. The workshop was facilitated by 4 professionals and attended by 15 members, who expressed their interest in participating after the sessions were convened through the network, including 8 scientists (two females), 5 representatives of fisher organizations (one female), and 1 female representative, each from a regional fisheries administration and an ENGO. Detailed minutes of the session were obtained,

reviewed by the core group after the session, and the results were synthesized in this paper by all of the co-authors (Figure 1).

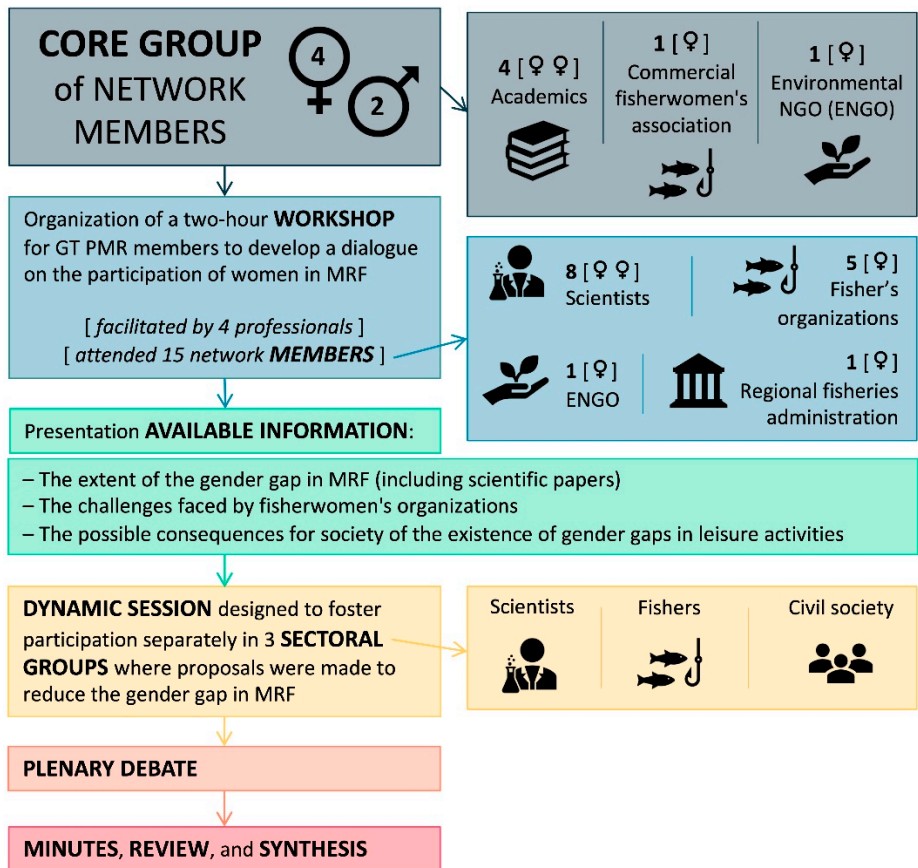

**Figure 1.** Diagram representing the workflow followed to address the consequences of the gender gap in marine recreational fisheries.

Freshwater recreational fishing (FRF) differs substantially from MRF. It is common for recreational fishers to specialize in one of the two fisheries, and they are usually ruled separately, so they are mostly approached independently by academic researchers [23]. For this reason, the results of this work are limited exclusively to MRF, anticipating that the results of an equivalent diagnosis carried out on FRM would probably be different.

## 3. Results

### 3.1. How Wide Is the Gap?

Less than a third of total recreational fishers are women in countries like Spain [24] or the USA [25]. According to the limited available research, the participation of women in MRF is very low, even when compared with other outdoor recreational activities [24]. Recreational fisherwomen tend to be younger than fishermen and show a relatively higher willingness to increase their knowledge about MRF through contact with other fishers [24]. In fact, although individual motivations such as the search for relaxation in outdoor environments are also important [26], the main motivation of women that engage in MRF is to spend time with their families, also above catch-oriented motivations [24].

### 3.2. Why Mind the Gap?

The reasons that explain the low participation of women in MRF have been little explored. Most available research blames social barriers that women face in participating in leisure activities in general, which include safety reasons, violence, social conventions about the appropriateness of some leisure activities for women, and the sense of a lack of

entitlement to leisure, mainly because women, in general, have less free time because they are far more often assigned than men to tasks related to the home and family care [25]. In fact, women abandon MRF earlier than men [27].

Globally, women's access to, ownership of, and control of natural resources is typically more restricted than for men, creating a gender gap in resource governance, an unfair division of labor, and missed opportunities for implementing sustainability initiatives which women tend to drive, among other things [28,29]. The wage gap between women and men [30] could also partly explain the gender gap in MRF. Furthermore, the historical sexual division of labor could explain the lower incorporation of women into MRF too: the sea is a predominantly masculine domain [31], while traditionally, women remained on land, shell-fishing, repairing nets, or processing seafood [32]. According to maritime culture and tradition, and in the derived collective imagination, women on board boats were a source of bad luck [33]. This would explain the difficulties faced by fisherwomen's organizations to move towards gender parity in fisheries governance, which results in the low representation of women, especially in management positions, even in fisheries where women are the majority [34]. In countries like Spain, with the largest number of women working in the EU fisheries sector, especially within shell-fishing by foot (contrary to shell-fishing from boats, traditionally carried out by men), there is a strong decline in female employment due to regulatory processes and a progressive masculinization of the sector related to economic profitability and social prestige [35]. This systematic relativization of the role of women in fishing has deep economic and social consequences, including impacts on women's quality of life [36].

Whatever the causes of the low participation of women in MRF, it is essential that democratic societies promote equality of opportunities between genders [37]. We propose that women's participation in MRF should include full access to the benefits derived from the practice of leisure activities, such as being able to freely and safely participate and compete and build careers in sport and physical activity, as advocated by the Brighton Declaration on women and sport [38].

### 3.3. A Roadmap to Gender Equality in Recreational Fisheries

After the participatory session at the GT PMR workshop, four key actions were agreed upon to begin to reduce the gender gap in MRF, as we show in Figure 2.

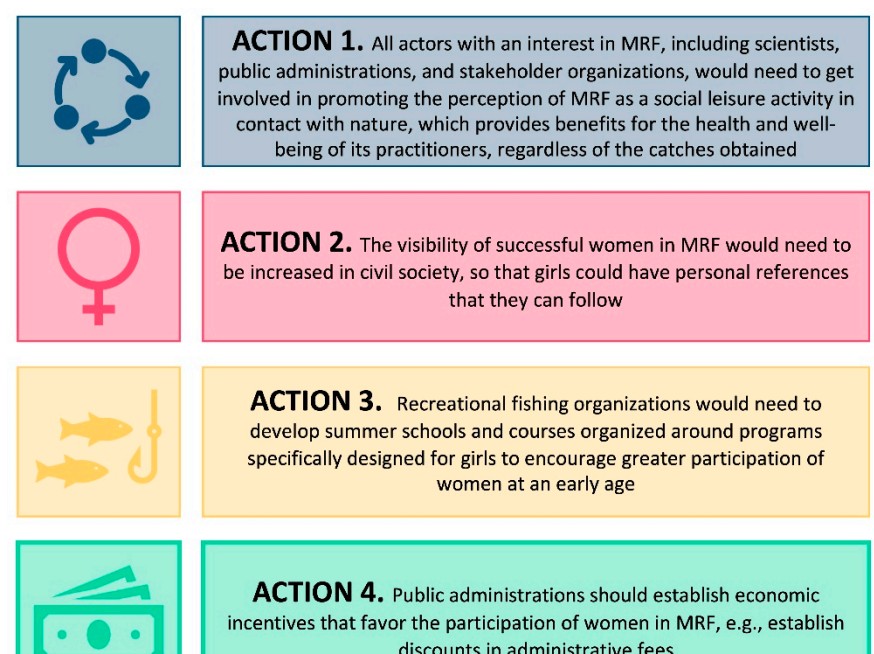

**Figure 2.** Key actions of the agreed roadmap to gender equality in marine recreational fisheries.

## 4. Discussion

This work is largely based on opinions expressed by the attendees of the workshop organized by the GT PMR. Views from scientists were mainly built from their research, while the contribution of the representatives of fishers' associations were largely determined by first-hand practical knowledge. Despite it being difficult to perform generalizations from this type of information, since our results as derived from a wide selection of key informants with a high degree of knowledge about different aspects related to the involvement of women in different fisheries, it is expected that they will be a useful guide for research and governance actions to address the gender gap in MRF.

Reducing the gender gap in MRF could support increased access to natural resources for women. However, an unintended consequence derived from the growth of the participation of women in MRF that must be considered is the potential added pressures on marine ecosystems exerted by both veteran fishers and newcomers. Recreational fisherwomen tend to exhibit higher awareness about animal welfare and environmental conservation compared to fishermen [24], which could contribute to reducing the negative impacts on fish stocks derived from the increased participation of women if they were enabled to fish in ways that coincide with these values. However, it seems that angling women retain more fish, mainly large fish with higher reproductive potential for consumption than men, who release more legal fish, including large ones, and are more supportive of regulations limiting efforts [8].

Recreational fishers tend to adopt attitudes that reduce fishing mortality as their involvement in fishing increases [39]. Consequently, behaviors that reduce negative impacts, such as limiting fishing mortality, would grow as newcomer's experience in fishing increases (both women and men). This is why the first of the actions we proposed in the roadmap to address the gap in MRF (Action 1) highlights the benefits derived from fishing, rather than focusing on catches. It would also be important in relation to the proposal to develop programs that encourage participation in MRF (Action 3) which include specific training aimed at reducing the negative impacts of fishing.

The development of the roadmap that we propose to reduce gender inequality in MRF will contribute specifically to Objective 5 (Gender Equality) of the United Nations (UN) Sustainable Development Goals (SDGs). Furthermore, Action 1 is part of the practice of responsible recreational fishing, which is consistent with Goals 2 (Zero Hunger), 3 (Good Health and Well-being), and 14 (Life Below Water). In this way, the consumption of the recreational catches, even if it does not represent a significant part of the diet, can contribute to the food security in a way that is compatible with the development of commercial fisheries, since recreational fishers tend to buy more and better quality fish in local markets [40]. In relation to Goal 3, the practice of MRF has been associated with benefits for the health and well-being of people, which has implications for public health systems, especially since it could favor the active aging of fishers, with the elderly in a high proportion [10]. Finally, focusing on the overall fishing experience, rather than maximizing catches, will reduce pressure on targeted fish stocks by recreational fishers, especially for newcomers [41].

We expect that the alignment of our roadmap to reduce the gender gap in MRF with international policies to promote sustainable development will increase the number of opportunities to develop the strategy. For instance, it could strengthen decision-makers to establish economic incentives to increase women's participation in MRF (Action 4). Fortunately, there are a growing number of organizations that are contributing to giving visibility to women's place at sea [42]. In a country like Spain, with relatively high female participation in some commercial fisheries, organizations such as the National Association of Women in Fisheries [43] and the Catalan Association of Women of the Sea [44] have been playing an important role by supporting fisherwomen and promoting their participation in blue economy business projects, providing female references to the next generations (Action 2). International organizations like the Women's Fishing Association contribute to

these same goals in MRF [45]. Expanding female leadership will undoubtedly contribute to increasing female participation in MRF [46].

## 5. Future Directions

Increasing the participation of women is a major social shift in terms of the way MRF is carried out, which needs the involvement of some women in the lead. Although the four actions we suggest are only a starting point, they could require some time to be accepted and fully implemented. There may be resistance among some fishermen to this change, who may not support different values and behaviors of fisherwomen, especially in decision-making, as has happened in other sports activities [47]. However, it is urgent and necessary that the gender gap in MRF is included in the agendas of scientists, fishery managers, policy-makers, stakeholder organizations, and civil society in general because gender-related issues in leisure activities have been neglected for too long [48].

Scientists need to increase scientific knowledge on the gender-related impacts on ecosystems and coastal communities to guide appropriate and unbiased fisheries governance. To ensure that fishery stocks and ecosystems are managed equitably and sustainably, fisheries managers need to develop adaptive data collection frameworks that provide updated information on the involvement of both men and women in fisheries that feed back into academic research. For their part, public administrations and stakeholders should cooperate to ensure that the distribution of benefits derived from marine ecosystem services is gender equitable.

**Author Contributions:** Conceptualization, P.P. and S.V.; investigation, J.S. and P.P.; writing—original draft preparation, J.S., P.P. and S.V.; writing–review and editing, all authors; project administration, A.T., P.P. and S.V.; funding acquisition, P.P. and S.V. All authors have read and agreed to the published version of the manuscript.

**Funding:** This work received funds from the project Grupo de Trabajo en Pesca Marítima Recreativa (GT PMR), funded by the Fundación Biodiversidad of the Spanish Ministerio Para la Transición Ecológica y el Reto Demográfico, co-funded by the European Maritime and Fisheries Fund and from Xunta de Galicia under the modality of Grupos de Referencia Competitiva (Grant ED431C2019/11).

**Institutional Review Board Statement:** Not applicable.

**Informed Consent Statement:** Not applicable.

**Data Availability Statement:** No data was generated in this study.

**Acknowledgments:** We appreciate the involvement of all experts who shared their knowledge during this study, largely facilitated by members of the Grupo de Trabajo en Pesca Marítima Recreativa of Spain (GT PMR). We also thank Altekio and ICSEM for their help in organizing and facilitating the workshop and the participatory sessions of the GT PMR.

**Conflicts of Interest:** The authors declare no conflict of interest.

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
