# Peer review of "Mind the Gender Gap in Marine Recreational Fisheries"

_sustainability, doi:10.3390/su151411292_

Round 1
Reviewer 1 Report
This opinion letter touches on an important issue of gender inequality in marine recreational fisheries. The article is easy to read and logical. Recommendations are not new, but I think are still valuable and applicable to the industry. To improve the contribution to the literature, I suggest the authors should expand the discussion on the actions proposed in the roadmap section. It would benefit readers if you could provide more in-depth discussion and information about the roadmaps.
Author Response
Q1: This opinion letter touches on an important issue of gender inequality in marine recreational fisheries. The article is easy to read and logical. Recommendations are not new, but I think are still valuable and applicable to the industry. To improve the contribution to the literature, I suggest the authors should expand the discussion on the actions proposed in the roadmap section. It would benefit readers if you could provide more in-depth discussion and information about the roadmaps.
Answer: We appreciate the positive feedback from reviewer #1. As it was suggested, we have expanded the discussion on the actions integrating the proposed roadmap to reduce the gender gap in Marine Recreational Fishing (MRF). Please, see lines 199-216, where we explain how the strategy we propose is aligned with international policies, which will benefit the development of the different actions included in the roadmap.
Reviewer 2 Report
I have questions regarding the paper for its improvement and they are in the annotated pdf form attached. That includes whether the increased access of women to the MRF will actually increase sustainability, and whether this increased gender participation is also the same as dive tourism done in some tropical islands that collect shells, pearls or large-bodied fish

I think the English is already acceptable
Author Response
I have questions regarding the paper for its improvement and they are in the annotated pdf form attached. That includes whether the increased access of women to the MRF will actually increase sustainability, and whether this increased gender participation is also the same as dive tourism done in some tropical islands that collect shells, pearls or large-bodied fish.
Answer: We appreciate the positive feedback from reviewer #2. We refer to the different suggestions in the annotated pdf:
Q1: Why is this relevant to mind the gender gap in the recreational fisheries, but how about in the commercial or small-scale fisheries in Spain? what's so special with the recreational fisheries?
Answer: We agree with the reviewer in that many (or most) worldwide fisheries show relevant gender gaps, in general showing more participation from men, although there are some fisheries mostly developed by women (e.g., many shellfisheries). Please, see that we have included some justifications in the abstract for our focus on recreational fisheries. Please, see lines 26-28.
Q2: But why are women excluded, are they not perhaps because they would be vulnerable socially and physically when they are included in these sports or even jobs?
Answer: Yes, we have reworded the sentence, as suggested, to improve the readability. Please, see lines 63-66.
Q3: What are the possible negative consequences of the gender gap for recreational and commercial fisheries?
Answer: Please, note that we explained about potential consequences in lines 63-66. We have stressed the need for information, as suggested, in lines 70-73.
Q4: What can women contribute that will differentiate them when it comes to recreational fisheries? Are there not other examples from publication or even tourism sector such as diving?
Answer: We are not aware of significant examples in other aquatic sports, like diving. However, we have now reworded the sentence to justify why our assessment was needed considering previous studies. Please, see lines 80-81.
Q5: This one needs to be defined again in this section since it is its first mention.
Answer: Please, note that we defined it in line 68, very close to this first mention in the methods section (line 87). We believe that it is unnecessary and redundant for readers to define commonly used terms in the text.
Q6: This one also needs to be defined again in this section since it is its first mention.
Answer: Please, see our previous answer.
Q7: Why are the numbers of the network only limited to 15 participants?
Answer: The sessions were open to all participants in the network. We have explained that in lines 101-102, as suggested.
Q8: The motivation is quite shallow if this is just spending time with family, are there not other motivations aside from leisure time? Do recreational fishers carry the fish with them too? Or is this catch and release kind of recreation?
Answer: In fact, it is their main motivation, even above catch-oriented motivations (including catch and release, and retentions). Please, see that we reworded in lines 124-125.
Q9: Badluck?? How does that explain advertisements showing women with men in yachts and getting a swordfish with their pack of cigarettes?
Answer: We are sorry, but we are not aware of these advertisements (we searched for them), and therefore we cannot give a reasonable feed back to the reviewer. In any case, we are quite sure about the traditional negativity derived from the female presence on board different types of boats, and especially on fishing boats.
Q10: Shellfishing or gleaning is not comparable to recreational fishing, perhaps diving could be more related because they could dive and see corals or catch some large fish using traps or nets but it is also hardly recreational in the traditional sense of the word of doing it as a leisure activity until you get your boat filled with tuna or swordfish.
Answer: We established comparisons with commercial fisheries because they both operate on marine living resources. We also found useful including some information on the governance on fisheries where female participation is dominant. Free or SCUBA diving could be seen as similar to recreational spear fishing (at least in terms of the materials used), but the numbers of spear fishers are much lower than anglers, while recreational fishers using gears like traps and nets are few, since these kinds of gears are only allowed in a small number of countries. We have not included these thoughts in the text, because we feel that would not be helpful for readers, but we are open to elaborate more and include some of these ideas if the reviewer find it useful.
Q11: Will the increased access or participation of women make the MRF inviting for them? Does this lead to more sustainable fisheries in the long run or the increased access lead to more catch/effort?
Answer: Please, note that we discuss about ecological effects derived from an increase in female participation in the discussion (lines 180-197). And we believe that more female leadership will drive more female participation in MRF, as we now included in lines 223-224.
Reviewer 3 Report
This paper provide an important but easily overlooked opinion. The authors focus on the gender gap in marine recreational fisheries and make some recommendations. I think this study will attract the attention of fisheries scientists and guide institutional activities and public policies in the future.
Author Response
This paper provide an important but easily overlooked opinion. The authors focus on the gender gap in marine recreational fisheries and make some recommendations. I think this study will attract the attention of fisheries scientists and guide institutional activities and public policies in the future.
Answer: We are glad that the reviewer #3 finds useful our study.
Reviewer 4 Report
Dear Authors,
Although improvements were made in the paper with the revision, the graphics expressing the situation were not included in the paper. In addition, no statistical comparison has been made between the gender gap.
In this state, it is seen that the study consists of only a report containing action plans. There is nothing interesting or impressive in the paper. Therefore, it would be more appropriate to make the paper more attractive with graphics.

Author Response
Dear Authors,
Although improvements were made in the paper with the revision, the graphics expressing the situation were not included in the paper. In addition, no statistical comparison has been made between the gender gap.
In this state, it is seen that the study consists of only a report containing action plans. There is nothing interesting or impressive in the paper. Therefore, it would be more appropriate to make the paper more attractive with graphics.
Answer: We appreciate the suggestions raised by reviewer #4. We refer to the different suggestions in the annotated pdf. Please, also note that we addressed the errors in the references.
Q1: This paragraph should be given in the material and method section.
Answer: The paragraph is in the Method section.
Q2: In the study, it would be beneficial to present the workshop results with graphics. The results of the gender gap in MRF should also be interpreted statistically within the scope of scientific articles.
Answer: We appreciate the suggestions on including some graphics showing the results of our study. Please, note that we describe in Figure 1 the procedure of our scientific design and methodology. We believe that readers will find this helpful to follow the text. In addition, we present our roadmap in Figure 2, as suggested. We also find that readers will find the text of the article more attractive in this way. Moreover, we find that there is no room to establish quantitative comparisons that could be analyzed by statistical methods. A meta-analysis would be difficult because the references we incorporated are few and often they are qualitative, and when they do include quantitative data, they are using different units, referring to different issues, etc. Please, take in mind that the few data available do not arise from a clinical analysis with a standard experimental design, but rather from a qualitative approach based on interviews.
Q3: Discussion section is written quite short and should be expanded a little more.
Answer: We have expanded the text in this section. Please, see lines 199-216.
Q4: The results and discussion sections should parallel each other and should not be disconnected from each other.
Answer: We structured the discussion to contextualize our findings and to anticipate some of the effects of the implementation of our roadmap to reduce the gender gap in MRF. Please, note that we have restructured the section to increase its coherence by introducing the pros and cons of our study first (see lines 170-177).
Round 2
Reviewer 4 Report
It is seen that necessary corrections and suggestions have been made in the revised paper. Thank you very much for your effort.
Best wishes